# Radiomics Analysis of Breast MRI to Predict Oncotype Dx Recurrence Score: Systematic Review

**DOI:** 10.3390/diagnostics15091054

**Published:** 2025-04-22

**Authors:** Nathan Kim, Richard Adam, Takouhie Maldjian, Tim Q. Duong

**Affiliations:** 1Department of Radiology, Montefiore Health System and Albert Einstein College of Medicine, Bronx, NY 10461, USA; nathan.kim@einsteinmed.edu (N.K.); tamaldjian@montefiore.org (T.M.); 2Medical School, New York Medical College, Valhalla, NY 10595, USA; radam39000@gmail.com

**Keywords:** breast cancer, magnetic resonance imaging, Oncotype DX recurrence score (ODXRS), radiomics, machine learning, dynamic contrast-enhanced (DCE) MRI, diffusion-weighted imaging (DWI), neoadjuvant chemotherapy

## Abstract

**Background/Objectives**: The Oncotype DX recurrence score (ODXRS) has emerged as an important tool for predicting recurrence risk and guiding treatment decisions in estrogen receptor-positive, human epidermal growth factor receptor 2-negative early-stage breast cancer. This review summarizes the current evidence on the clinical utility of the Oncotype DX RS and explores emerging research on potential imaging-based alternatives. The 21-gene assay provides a recurrence score that stratifies patients into low, intermediate, and high-risk groups, helping to identify patients who may benefit from adjuvant chemotherapy. Multiple validation studies have demonstrated the prognostic and predictive value of the ODXRS. However, the test is costly and requires tumor tissue samples. **Methods**: This paper systemically reviewed the current literature on the use of radiomic analysis of breast MRI to predict Oncotype DX. The literature search was performed from 2016 to 2024 using PubMed. We compared different image types, methods of analysis, sample size, numbers of high/intermediate and low scores, MRI image types, performance indices, among others. We also discussed lessons learned and suggested future research directions. **Results**: Recent studies have investigated the potential of radiomics applied to breast MRI to non-invasively predict the Oncotype DX RS. Quantitative imaging features extracted from dynamic contrast-enhanced MRI, diffusion-weighted imaging, and T2-weighted sequences have shown promise for distinguishing between low and high RS groups. Multiparametric MRI-based models integrating multiple sequences have achieved the highest performance. **Conclusions**: While further validation is needed, MRI radiomics may offer a non-invasive, cost-effective alternative for assessing recurrence risk.

## 1. Background

Breast cancer is a heterogeneous disease with varying prognoses and treatment responses. In recent years, gene expression profiling has emerged as a useful tool for assessing recurrence risk and guiding treatment decisions. Oncotype DX is a genomic test that plays a crucial role in the management of early-stage breast cancer, specifically for patients with ER+ (estrogen receptor positive), HER2− (human epidermal growth factor receptor 2 negative) breast cancer [1,2]. The test analyzes the expression of 21 genes from a sample of the patient’s tumor tissue, which is typically obtained through surgery or a biopsy [1,2]. Based on the activity levels of these genes, the test generates as Recurrence Score (RS) ranging from 0 to 100. The ODXRS is typically categorized into low (<18)-, intermediate (18–30)-, and high (greater than or equal to 31)-risk groups, although different studies use different thresholds [3]. This score predicts the risk of cancer recurrence within ten years after the initial diagnosis, as well as the potential benefit of adding chemotherapy to hormone therapy. Oncotype DX helps tailor breast cancer treatment to the genetic profile of the tumor, aligning with the principles of personalized medicine. The test’s ability to accurately stratify recurrence risk helps to avoid overtreatment, sparing patients from the side effects of unnecessary chemotherapy and improving their quality of life [4]. Improved survival outcomes have been reported using Oncotype DX [5]. It has also been successfully used for risk stratification to guide omission of radiotherapy in low-risk patients opting for breast conservation [6,7].

Oncotype DX has limitations. The test is invasive, requiring a tissue sample, and can be expensive, which may limit accessibility for some patients [1,2]. Additionally, it may be less prognostically accurate in Black women [8]. As a result, there has been research exploring non-invasive alternatives that could provide similar prognostic information. Of interest, a recent study attempting to correlate Ki-67 proliferative index with Oncotype Dx recurrence score found no correlation between these two parameters in a cohort of ER+, HER2− patients with early breast cancer [9].

Magnetic Resonance Imaging (MRI) is a valuable tool in the detection and management of breast cancer due to its ability to provide highly detailed images of breast tissue. Radiomic MRI signatures can allow prediction of recurrence free survival [10]. MRI Radiomic texture analysis also improves prediction of pCR [11]. Machine learning radiomic models can help improve breast cancer diagnosis and treatment [12]. Radiomics involves the high-throughput extraction of quantitative features from medical images. Despite the growing promise of radiomic analysis to predict Oncotype DX, there is a critical gap in our understanding: the lack of a systematic comparison between radiomics-based approaches and established genomic assays like Oncotype DX. This gap hinders our ability to fully evaluate the potential of radiomics as a complementary or alternative tool for recurrence risk assessment. To our knowledge there is currently no review that systematically compares different radiomics studies to Oncotype DX. Such a comprehensive review would be crucial in providing a clearer picture of the relative strengths and limitations of each method, guide future research efforts in developing better risk assessment tools and inform clinical decision-making and potentially improve patient outcomes. By addressing this gap, this review can pave the way for more personalized and effective treatment strategies in breast cancer management.

The goal of this study was to review the current literature on the use of radiomic analysis of breast MRI to predict Oncotype DX. We compared different image types, methods of analysis, sample size, numbers of high/intermediate and low scores, MRI image types, performance indices, among others. We also discussed lessons learned, challenges of deployment in clinical practice and suggested future research directions.

## 2. Materials and Methods

No ethics committee approval was required for this systematic review.

### Search Strategy and Eligibility Criteria

This review followed the PRISMA guidelines. The literature search was performed from 2016 to 2024 using the following key words: “breast cancer (oncotype [title])”, “Breast MRI oncotype DX RS score”, “Breast MRI recurrence score”, and “breast MRI oncotype dx assay”. The database was PubMed, although many of the journals were found on sites other than PubMed. Only original articles written in English were selected. The search and initial screening for eligibility were performed by NK and independently verified by TM and/or TD. A PRISMA flowchart was used to document the identification, screening, eligibility, and inclusion stages of study selection (Figure 1).

## 3. Results

PubMed search yielded 246, 9, 83, and 18 articles, respectively. After exclusion, a total of 16 articles were included in our study (Table 1). In this review, we first summarized individual papers and followed it with a generalization of lessons learned. We then discussed challenges of Oncotype DX and suggested future research directions.

## 4. Discussion and Conclusions

### 4.1. Summary of Individual Papers

#### 4.1.1. Texture/Radiomic Features

Li et al. explored the potential of radiomics in evaluating breast cancer recurrence risk by comparing computer-extracted breast MRI phenotypes with multigene assays: MammaPrint, Oncotype DX, and PAM50 [13]. An IRB-approved retrospective analysis of 84 breast MR examinations from four institutions from an imaging archive for which clinical, histopathologic, and genomic data was also available. The dataset comprised 74 (88%) ductal, 8 (10%) lobular, and 2 (2%) mixed cancers [13]. A total of 73 (87%) were ER+, 67 (80%) were PR+, and 19 (23%) were HER2+. Computer-extracted image phenotypes were compared to genomic assays for ability to assess for risk of breast cancer recurrence and ascertain predictive ability of MRI. After consensus by radiologists for tumor location, automated segmentation was performed, and 38 tumor phenotypes were automatically extracted from MR images. These phenotypes involved features related to tumor size, shape, margin morphology, enhancement texture, enhancement kinetics, and enhancement-variance kinetics. Multiple linear regression analyses were performed and demonstrated significant correlations between radiomic signatures and recurrence scores from various multigene assays. In distinguishing between good and poor prognosis, compared to MammaPrint, Oncotype DX, PAM50 risk of relapse based on subtype and proliferation, PAM50 relapse risk based on subtype, radiomics achieved area under the ROC (AUC) values of 0.88, 0.76, 0.68, and 0.55, respectively.

Saha et al. investigated whether multivariate machine learning models of algorithmically assessed MRI features from breast cancer patients are associated with ODXRS [14]. Data was procured from pre-operative DCE-MRI images (1.5 T and 3 T) of 261 female breast cancer patients from a single institution with available ODXRS. A total of 529 features were extracted using a computer algorithm. Two machine learning models were developed to discriminate high ODXRS from intermediate and low, and high and intermediate ODXRS from low. High against low and intermediate ODXRS were predicted by the multivariate model with an area under the curve (AUC) of 0.77. Low against intermediate and high ODXRS were predicted with an AUC of 0.51. Thus, a moderate association between imaging and ODXRS was identified, and the evaluated current models currently do not justify replacing ODXRS with imaging alone.

Nam et al. evaluated the ability of a radiomics signature (Rad-score) generated from 158 radiomic features (8 morphological, 76 histogram-based, and 72 higher-order texture features) extracted from DCE-MRI (3 T) to differentiate between low and non-low ODXRS risk groups in 67 female ER+ invasive breast cancer patients [15]. The least absolute shrinkage and selection operator (LASSO) was used to generate a radiomics signature (Rad-score) [15]. Patients were separated into low (<18) and non-low (≥18) risk groups based on ODXRS. The association between various clinicopathologic parameters, MRI findings, and Rad-score with ODXRS risk groups was evaluated by univariate and multivariate logistic regression analyses. The Rad-score for each tumor was derived from 10 (6.3%) of the 158 radiomics features. Non-low ODXRS risk group showed higher Ki-67, high p53, and higher Rad-score. Classification into low and non-low ODXRS risk categories based on Rad-score demonstrated an AUC of 0.759.

Chen et al. developed a multiparametric MRI-based radiomics model to assess the ODXRS in ER+/HER2− breast cancer patients [16]. A total of 151 ER+/HER2− breast cancer patients with 21-gene expression assays and pre-operative MRI were studied retrospectively to assess recurrence risk based on radiomics features. Patients were divided into training (n = 106) and validation (n = 45) cohorts. A total of 1045 radiomic features were extracted using manual lesion segmentation from T2WI, DWI, and DCE MRI (1.5 T or 3.0 T). Recursive feature elimination method for feature dimension reduction and the synthetic minority oversampling technique for dataset balance were employed. Linear support vector machine classifier models were built and tested for ability to differentiate high (greater than or equal to 26) from low (less than 26) ODXRS using T2WI, DWI apparent diffusion coefficient (ADC) maps, DCE individually and in combination (multiparametric), as well as a model for clinical data and a fusion model for clinical and multiparametric MRI data. DeLong’s test with Boferroni correlation were used. The AUC for multiparametric analysis was 0.92, DCE was 0.83, T2WI was 0.78, and ADC was 0.77 in the training group. The fusion model demonstrated AUC of 0.92 in training group and 0.78 in validation group, which was significantly better than the clinical model with AUC of 0.64 (training) and 0.62 (validation). Multiparametric MRI radiomics showed potential as a viable alternative to the Oncotype DX.

Chiacchiaretta et al. evaluated the potential of MRI radiomics to predict breast cancer recurrence risk based on features extracted from tumor and peritumoral tissues [17]. A total of 62 ER+/HER2− breast cancer patients with pre-treatment MRI and ODX scores were studied. ODXRS score of 25 was used as a cut-off between low/intermediate and high-risk categories. Two readers segmented the tumors. Partial lesion square (PLS) regression multivariate machine learning algorithm was employed. Largest magnitude (top 5%) of Beta-weights of radiomic features were included. Hyperparameter optimization and evaluation of generalizable performance was accomplished with leave-one-out nested cross validation (nCV). Complete dataset exploratory analysis demonstrated average absolute correlation among features of 0.51. nCV framework AUC was 0.76. In only tumor (T) or tumor plus surrounding tissue (TST), combining “early” and “peak” DCE images revealed that the TST approach tended toward statistical significance with an AUC of 0.61. The top 5% of features (47 in total) were balanced between T and TST (23 and 24, respectively), with 33/47 (70%) being texture-related and 25/47 (53%) derived from high-resolution (1 mm) images. This radiomics-based machine learning approach shows potential for accurately predicting recurrence risk in early ER+/HER2− breast cancer patients.

Romeo et al. used radiomics features from DCE-MRI and a machine learning (ML) algorithm to predict ODXRS in patients with ER+/HER2− invasive breast cancer [18]. ODXRS was considered positive if over 16 for patients under 50 years and over 26 for those over 50. Tumor lesions were manually annotated, and radiomic features were extracted from 3D ROI positions. Out of 248 patients, 87 had positive ODXRS. A logistic regression ML classifier, trained on 166 patients and tested on 82, was used for prediction. Of the 1288 features extracted, 92 were selected. The ML classifier achieved an accuracy of 63% in the test set, with 80% sensitivity, 43% specificity, and an AUC of 66%. The study suggests that radiomics and ML applied to DCE-MRI could potentially predict ODXRS non-invasively in breast cancer patients.

Arefan et al. determined whether quantitative background parenchymal enhancement (BPE) measurements from DCE-MRI in one or both breasts could predict recurrence risk in women with breast cancer using ODXRS as the reference standard [19]. This was a Health Insurance Portability and Accountability Act of 1996 (HIPAA)-compliant retrospective study from a single institution. Methodology involved a development set (1-2007 to 1-2012) and internal test set (1-2012 to 1-2017) from a single institution, quantitative BPE computed using and in-house algorithm in both breasts, univariable logistic regression analyses, and ODXRS divided into low/intermediate (less than or equal to 25) and high risk (over 25) groups. A total of 127 women with a mean age of 58 were classified as 33 in high risk and 94 and low/intermediate risk based on ODXRS. A total of 20 of the 127 demonstrated local or distant recurrence, for a recurrence rate of 15.7% at least 10 years of follow-up. The test set comprised 60 women with a mean age of 57.8, of which 16 were high risk and 44 were low/intermediate risk. BPE was associated with high risk ODXRS. BPE combined with tumor radiomics demonstrated an AUC of 0.94 in the development set and 0.79 in the test set for differentiating low/intermediate from high risk ODXRS. A negative predictive value for local or distant recurrence of 0.97 and 0.93 was achieved. Ipsilateral and contralateral DCE MRI measurements of BPE can differentiate high from low/intermediate recurrence risk.

Fan et al. identified an association of radiogenomic signatures with ODXRS, and assessed these signatures for feasibility as biomarkers of survival and response to neoadjuvant chemotherapy (NAC) in patients with ER+ breast cancer [20]. This retrospective study involved three datasets. The development dataset (130 women; mean age 52) of pre-operative DCE-MRI and ODXRS 1-2016 to 10-2019 identified the radiogenomic signatures. The prognostic assessment dataset (135 women; mean age 50) evaluated the prognostic significance of imaging signatures regarding overall and recurrence-free (RFS) survival utilizing multivariable Cox proportional hazards model (116 women; mean age 48). The treatment dataset (8-2015 to 3-2019) evaluated therapeutic significance of radiogenomic signatures in terms of predicting response to NAC. AUC was calculated for prediction performance. Textural, morphological, and statistical features were identified in 11 radiogenomic signatures. An ODXRS score of 29.9 or greater signified poor overall survival and RFS. Combining predicted ODXRS with radiogenomic signatures improved response prediction and can be helpful for prognosis and treatment.

#### 4.1.2. Radiologist-Defined MRI Features

Kim et al. evaluated whether MRI features could be used to predict ODXRS in patients with ER+/HER2− invasive breast cancer [21]. A total of 473 ER+/HER2− invasive breast cancer patients (with a total of 485 cancers) with pre-operative MRI and Oncotype DX assay were included. Low ODXRS (less than 18) scores were represented by 288 of 485 (59.4%) cancers, intermediate ODXRS (18–30) scores by 155 of 485 (31.9%) cancers, and high ODXRS (>31) scores by 42 of 485 (8.7%). Independent predictors of low and high ODXRS based on logistic regression analyses of MRI review demonstrated that a round shape and low proportion of washout component were associated with low ODXRS. Heterogeneously dense (OR = 3.205) or scattered fibroglandular breast tissue, a non-spiculated margin, and low proportion of persistent component were associated with high ODXRS. MRI features showed the potential for discrimination of ODXRS in patients with ER+/HER2− invasive breast cancer.

Galati et al. combined breast MRI-derived biomarkers with clinical-pathological parameters to identify patients most in need of the Oncotype DX Breast Recurrence Score (ODXRS) assay, which predicts the benefit of adjuvant chemotherapy in ER+/HER2− early breast cancer [22]. In this retrospective study of 58 patients, patients who had pre-operative multiparametric breast MRI at 3 T and ODXRS profiling were included. Two radiologists evaluated the imaging using the ACR BI-RADS lexicon. Results showed positive correlations between ODXRS and tumor size, stage, grade, and BI-RADS categories, suggesting that BI-RADS categories can predict therapeutic decisions. This study indicates that a combined onco-radiologic assessment could effectively predict decisions typically based on ODXRS results, potentially personalizing treatment and reducing healthcare costs.

Dialani et al. reviewed mammographic (MMG), ultrasonographic (US), and MRI features and pathologic characteristics of ER+/LN− invasive breast cancer to determine the relationship of these characteristics to Oncotype DX recurrence scores (ODXRS) for breast cancer recurrence [23]. This IRB-approved retrospective study included 319 patients with ER+/LN− invasive breast cancer who underwent genomic testing. MMG was performed in 86% of patients, US in 84%, and MRI in 33%, including morphologic and kinetic evaluation. Images from each imaging modality were evaluated. Each imaging finding, progesterone receptor (PR) and HER2 status, and tumor grade were then individually correlated with ODXRS. Analysis of variance determined differences for each imaging feature, and regression analysis calculated the prediction of recurrence based on imaging features combined with histopathologic features. Specific imaging features that showed correlation with ODXRS were mass shape at MMG (significant P value due to oval shape versus the other shapes), vascularity and posterior acoustics at US, and lesion shape on MRI (significant P value due to difference between lobulated versus other shapes). A predictive model for recurrence scores was developed using these imaging features in combination with PR and HER2 status and tumor grade, considering a score > 30 as a high recurrence score. PR+ trended with a lower recurrence score relative to PR− while HER2+ trended with a higher recurrence score relative to HER2−. Using regression tree analysis, there was a correlation with 89% sensitivity and 83% specificity. Based on preliminary data, information obtained routinely for breast cancer diagnosis can reliably be used to predict the ODXRS with high sensitivity and specificity.

Tsukada et al. evaluated whether pre-operative imaging techniques (MMG, US, DCE-MRI, and PET/CT) could predict Oncotype DX Breast Recurrence Score (ODXRS) in ER+/HER2− early-stage breast cancer, aiming to reduce reliance on the costly test [24]. This retrospective study included 51 patients, categorizing ODXRS as low (<18) or intermediate/high (≥18) [24]. Tumor characteristics from imaging were analyzed, and predictors of low ODXRS were identified using logistic regression. Univariate analysis found nuclear grade, tumor margin, tumor growth orientation on MRI, and SUVmax on PET/CT were significantly linked to low ODXRS. Multivariate analysis showed that tumor growth orientation perpendicular to Cooper’s ligament on MRI and low SUVmax on PET/CT were independent predictors of low ODXRS. A SUVmax cut-off of 3.0 had high sensitivity (94.4%) and specificity (73.0%). These findings suggest that MRI and PET/CT features can predict low ODXRS, potentially identifying patients who may not need the expensive test, thus reducing healthcare costs.

#### 4.1.3. Other Methods

Kim et al. evaluated MRI from 105 ER+/HER2− node negative patients’ ADC values from whole-lesion histograms (pixel-based whole tumor analysis) and breast cancer recurrence risk [25]. ADC histogram features were extracted. Intratumoral heterogeneity was assessed based on computed ADC difference value (derived from difference between 5th and 95th percentiles of ADC). Quantitative ADC and recurrence risk based on ODXRS (low risk < 18; non-low risk less than or equal to 18) were correlated. A multivariate regression analysis revealed that small tumor size and lower ADC difference value were associated with low recurrence risk, implying that the ADC difference value obtained from the whole lesion histogram may be a viable quantitative DWI biomarker for recurrence risk. This approach might help discriminate between low-risk and non-low-risk recurrence groups and provide insight into intratumoral heterogeneity.

Thakur et al. measured the apparent diffusion coefficient (ADC) values in ER+/LN− invasive breast cancer and investigated the correlation of ADC with ODXRS [26]. This retrospective study included 31 patients who underwent pre-operative 3.0T MRI scans with DWI sequences and also ODXRS genomic testing [26]. ADC_600_ and ADC_1000_ parametric maps were generated, and ADC values were calculated from a user-drawn region of interest. ODXRS separated 10-year recurrence risk as into low (<18), intermediate (18–30), or high (>30) categories. All breast lesions, including subgroups of invasive ductal carcinoma (IDC) lesions and mass-only lesions, were dichotomized by ODXRS scores, low-risk versus intermediate/high-risk, and statistical analysis included Mann–Whitney’s test, receiver operating characteristic (ROC) curves, and multivariate analysis. Invasive breast cancers, when scored as low-risk by ODXRS, had significantly higher ADC values compared to intermediate/high-risk lesions for both ADC_600_ and ADC_1000_ mean values for both only mass-lesions or only IDCs. Preliminary findings suggest that lesion ADC values correlate with recurrence risk likelihood as stratified using ODXRS. Thus, ADC could potentially serve as a surrogate biomarker for tumor aggressiveness.

Zhang et al. investigated the relationship between ODXRS and contralateral non-tumor breast MRI BPE in 80 women (mean age 51.1) [27]. Study group was comprised of ER+/HER2− early-stage invasive ductal carcinoma with breast presurgical MRI, ODXRS scoring (46 patients low risk less than or equal to 17; 34 patients intermediate/high risk > 17), and breast conservation surgery (2008–2010). Automatic extraction of BPE by k-means clustering from pre-contrast and three post-contrast phases. Initial enhancement (IE), late enhancement, overall enhancement (OE), and AUC were computed. First-order metrics were determined from histogram analysis and comparison to ODXRS was performed using Mann–Whitney tests and Spearman-rank correlation analysis. For the mean of the top 10% pixels, significant differences were noted for IE, OE, and AUC. Using the risk score as a continuous variable, correlation analysis showed a weak but significant correlation with the mean of the top 10% pixels for IE (r = 0.26), OE (r = 0.25), and AUC (r = 0.27). BPE enhancement metrics in the non-tumor breast are associated with tumor ODXRS, suggesting that the breast microenvironment may be related to the likelihood of recurrence and the potential benefit of chemotherapy. Thus, MRI-derived BPE metrics could potentially serve as imaging biomarkers for tumor aggressiveness and treatment response in ER+/HER2− early-stage breast cancer.

Ha et al. determined if convolutional neural networks (CNNs) could predict ODXRS using MRI data, potentially offering a non-invasive and less expensive alternative [28]. This IRB-approved retrospective study included 134 patients with ER+/HER2− invasive ductal carcinoma who underwent both breast MRI and ODXRS evaluation. Patients were classified into three groups: low risk (group 1, ODXRS < 18), intermediate risk (group 2, ODXRS 18–30), and high risk (group 3, ODXRS > 30). The study used 1.5 T and 3.0 T T1 postcontrast MRI images (1.5 T and 3 T), with 3D segmentation of breast tumors yielding 1649 volumetric slices from 134 tumors. A CNN consisting of four convolutional layers and max-pooling layers, with 50% dropout applied to the second to last fully connected layer to prevent overfitting, was employed. Two prediction models were performed: a three-class prediction model (group 1 vs. group 2 vs. group 3) and a two-class prediction model (group 1 vs. group 2/3). The three-class prediction achieved an overall accuracy of 81%, with specificity 90%, sensitivity 60%, and the AUC was 0.92. The two-class prediction performed slightly better, with an overall accuracy of 84%, specificity of 81%, sensitivity of 87%, and an AUC of 0.92. These findings demonstrate the feasibility of training deep CNNs to predict ODXRS.

### 4.2. Summary of Lessons Learned

The studies, which were primarily single center studies, originated mainly from the United States, Asia, and Europe. They were retrospective in nature, except for 4 of the studies. Sample sizes ranged from 31 to 473 patients. Samples sizes for high/intermediate ODXRS and low ODXRS were comparable and were not skewed toward a certain direction.

Multiple studies demonstrated that MRI-based radiomics features can predict ODXRS with moderate to good accuracy, implying that radiomics could replace or complement genomic testing as a non-invasive alternative. Radiomics analysis of MRI images to predict the ODXRS demonstrate that radiomics signatures derived from dynamic contrast-enhanced (DCE) MRI, T2-weighted imaging, and diffusion-weighted imaging (DWI) can be associated with ODXRS and recurrence risk [13,14,15,16,17,18,19,20]. Predictive features emerging repeatedly included tumor size, shape, margin morphology, enhancement texture, and kinetic assessment. A multiparametric approach combining features from different MRI sequences (T2-weighted, DWI, and DCE) generally improved predictive performance compared to using a single sequence alone [16].

The quantitative assessment of Background Parenchymal Enhancement (BPE) in both the affected and contralateral breasts has shown potential in predicting recurrence risk, with some studies finding correlations between BPE measures and ODXRS [27]. Several studies have explored the relationship between Apparent Diffusion Coefficient (ADC) values derived from DWI and ODXRS. Some have found that tumors with lower RS tend to have higher ADC values, suggesting that ADC could serve as a surrogate biomarker for tumor aggressiveness [26]. Measures of intratumoral heterogeneity, such as the ADC difference value (the difference between the 5th and 95th percentiles of ADC values within a tumor), have shown promise in distinguishing between low and non-low risk recurrence groups.

Various machine learning algorithms, including support vector machines, logistic regression, and convolutional neural networks, have shown promise in utilizing multiple radiomics features to predict ODXRS risk categories. Several studies also found that including radiomics features from the peritumoral region improves prediction accuracy, highlighting the role of the tumor microenvironment in recurrence risk. The quantitative assessment of BPE in both the affected and contralateral breasts was associated with recurrence risk, with some studies finding correlations between BPE measures and ODXRS. Multiple studies reported associations between ADC values from DWI and ODXRS, with lower ADC values generally correlating with higher recurrence risk, indicating that diffusion characteristics may reflect tumor aggressiveness. Some studies looked at predicting response to NAC [20]. Some studies were multiparametric in design while some studies were multimodality in nature [16,23,24]. Some studies used machine learning and some studies used deep learning. The highest AUC for MRI was obtained with a deep learning CNN model [28].

Regarding performance, the area under the ROC curve values for predicting ODXRS categories typically ranged from 0.75 to 0.92, indicating good to excellent discrimination.

### 4.3. Perspectives and Conclusions

New ways to assess Oncotype DX RS have the potential to completely change determining recurrence risk for breast cancer, leading to better patient outcomes. If validated, the radiomics approaches could potentially reduce the need for invasive, expensive genomic testing in some patients. The first key finding was that MRI-based radiomics features can predict ODXRS with moderate to good accuracy and good discrimination ability, and the second one was that multiparametric approaches combining features from different MRI sequences generally outperformed single-sequence models in predicting ODXRS risk categories. These findings suggest the use of MRI radiomics as a non-invasive alternative or to complement genomic testing for assessing breast cancer risk, possibly decreasing the dependence on invasive biopsies and expensive genetic assays. The ability to extract quantitative imaging biomarkers from routine MRI scans could potentially streamline clinical decision-making and improve accessibility to better prognostic tools.

However, there was considerable variability in imaging protocols, feature extraction methods, and analysis techniques across studies, emphasizing the need for standardization in radiomics research and ODXRS. Also, larger, multi-center studies are needed to further validate the performance of radiomics models across different patient populations and imaging equipment, and prospective studies are needed to directly assess the ability of imaging-based models to predict long-term clinical outcomes. The integration of radiomics with other data types (clinical, pathological, genomic) and the use of more advanced AI techniques may further improve predictive performance. MRI-based radiomics shows considerable promise as a non-invasive tool for assessing breast cancer recurrence risk. While it may not entirely replace genomic testing, it has the potential to serve as a complementary tool, potentially reducing the need for invasive biopsies and expensive genetic assays. As the field continues to evolve, the integration of radiomics with clinical, pathological, and genomic data may lead to more personalized and effective strategies for breast cancer management. This approach can be combined with other patient data and other deep learning models to predict outcomes, such as response to neoadjuvant chemotherapy and survival [29,30,31]. Future research should focus on addressing the challenges of standardization, validation, and clinical integration.

Sources of heterogeneity among studies may negatively impact the ability to achieve uniform results. Variables that contribute to heterogeneity include different scanners (manufacturers), differences in field strength, and variation in protocols and technical parameters. This can be particularly problematic when dealing with multi-institutional studies with different scanners and protocols, causing the study design to be flawed by lack of uniformity, potentially adversely affecting results. Addressing this can be challenging. Li et al. attempted to mitigate this problem in their study involving data from 4 institutions by limiting inclusion to patients scanned on 1.5 T GE Medical systems imaging units (Milwaukee, Wis). The process and methods of segmentation also introduce variability. While most studies utilized some method of automated segmentation for radiomics analysis, one study used purely radiologist drawn ROI, which is very work-intensive [15]. Sample size is an important factor impacting prognostic accuracy in radiomics. The extraction of a large number of features from a small sample size may lead to overfitting and limit generalizability. Only three of the studies included more than 300 patients [20,21,23]. Single institution studies also limit generalizability, and these comprise most of the studies in this review. While texture features can be useful, radiomics extract a broader range of data, including tumor shape, size, and post-contrast enhancement kinetics, which may improve prediction analyses. Building on the principle that more data can improve accuracy, multiparametric radiomics, which exploit the advantages of radiomics and include multiple types of image acquisition (DCE, T2, DWI), as might be expected, were shown to yield the highest AUC (92%) among the studies [16]. The use of a deep learning CNN algorithm was able to predict ODXR with a similarly high AUC of 92% [28].

Clinical application of radiomics models necessitates overcoming challenges across vast domains ranging from technologic issues to ethical concerns before widespread implementation can be contemplated. Lack of technical standardization stemming from variations in scanning protocols, scanners by different manufacturers, and variations in scanning technique can lead to data variability, undermining the optimization and application of a uniform radiomics model, as discussed. Beyond these inherent challenges to model generalizability, radiomics analysis is computationally intense, requiring vast resources to accomplish its tasks. A highly sophisticated infrastructure with high-capacity servers and high capacity, robust AI would be needed to support the feasibility of real-time radiomics implementation. Financial considerations are therefore a major factor, including the expenses of hardware/software acquisitions, computational power, and training of staff. These challenges present a formidable barrier to clinical implementation of radiomics models, most severely impacting resource-limited healthcare facilities in underserved areas with limited staffing where it is arguably needed the most. Another obstacle to the integration of radiomics into daily clinical workflow are the ethical concerns that arise with such implementation. Compliance with ethical standards, including privacy concerns/data sharing, risk of algorithm bias, and patient consent and informed decision-making issues, would need to be addressed. The “black box” nature of machine learning algorithms gives no explanation for how the algorithm arrives at a decision, making it difficult to establish trust with and gain acceptance from clinicians and patients. Strategies that address some of these challenges include strict adherence to data protection laws, strengthening generalizability through testing and validation of radiomics models across diverse datasets and clinical settings, and standardization of imaging protocols, model design, and feature extraction techniques.

Interdisciplinary collaboration between radiologists, oncologists, scientists, and ethicists can help to develop clinically effective radiomics solutions that are compliant with privacy laws and ethical standards. Transparency with patients by including them in the discussion about the use of their data and the benefits of radiomics models and ensuring data security and patients’ rights to privacy can promote trust and patient acceptance. Addressing these challenges would make the clinical application of radiomics a feasible and desirable resource to add to the clinician’s toolbox to improve diagnostic accuracy, promote personalized treatment plans, and ultimately improve patient outcomes.

## Figures and Tables

**Figure 1 diagnostics-15-01054-f001:**
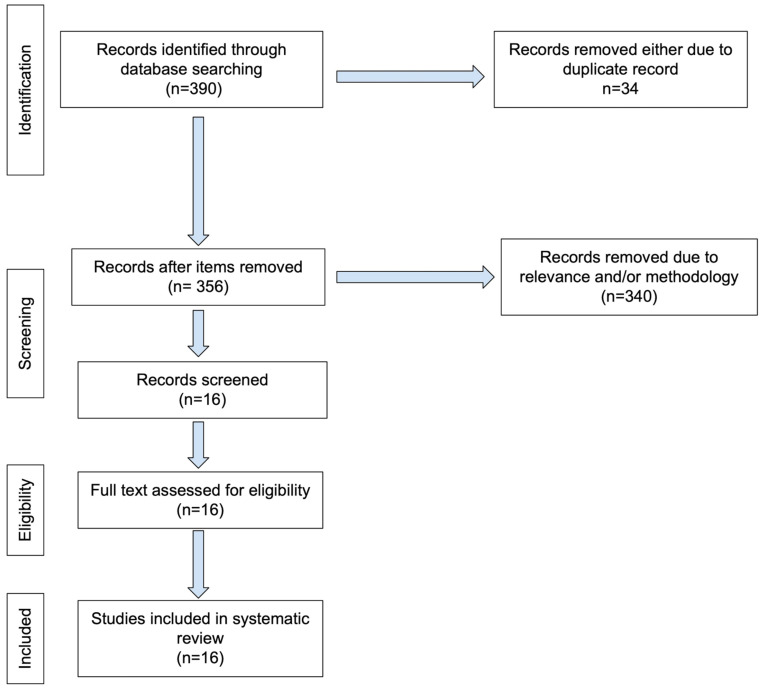
PRISMA flowchart.

**Table 1 diagnostics-15-01054-t001:** Summary of studies.

Study	Source	Duration	# of pts	High/Mid	Low	Image Type	Method	Classifier	Cross-Validation	AUC	Accu	Sens	Spec
Texture/radiomic features													
Li (2016) [13]	TCIA		84			DCE	Texture features	Logistic regression	LOOCV	88% (MammaPrint) 76% (ODX)			
Saha (2018) [14]	Duke	1 Jan 2000–23 Mar 2014	261	116	145	DCE	Texture features	Logistic regression	Training, independent	77% (high vs. mid/low) 51% (low vs. high/mid)			
Nam (2019) [15]	Republic of Korea	May 2011–Mar 2016	67	3, 19	45	DCE	Texture features	Logistic regression	LOOCV	75.90%			
Chen (2023) [16]	China	Apr 2017–Mar 2019	151 (T:106, V: 45)	88	63	T2WI, ADC, DCE	Radiomics features	Linear support vector machine		92% (multiparametric) 83% (DCE), 78% (T2WI), 77% (ADC)			
Chiacchiaretta (2023) [17]	Italy	Jan 2016–May 2020	62	15	47	DCE-MRI	Radiomics features (nCV, TST)	Partial least square (PLS) regression	nCV	76% (nCV) 61% (TST)			
Romeo (2023) [18]	Italy	1 Jan 2000–23 Mar 2014	248	87 (+)	161 (−)	Subtracted DCE	Radiomics features	Logistic regression	Hold-out	66%	63%	80%	45%
Arefan (2024) [19]	U of Pittsburgh	Jan 2007– Jan 2017	187 (T:127, V:60)	16	44	DCE-MRI	Tumor radiomics+ BPE	Linear discriminant analysis	10-fold	79%			
Fan (2022) [20]	China	Aug 2015–Oct 2019	381	118	12	DCE	Radiogenomic signatures (elastic net regression)	Multivariate Cox proportional hazards	10-fold	85%			
Radiologist-defined features													
Kim (2021) [21]	Republic of Korea	Jan 2015– Dec 2018	473	197	288	DCE	Radiologist-defined features; Washin/washout	Logistic regression			84%	52.2%	71%
Galati (2022) [22]	Italy	Apr 2017– Jan 2018	58	23	35	T2WI, ADC, DCE	Radiologist-defined MRI features	Logistic regression					
Dialani (2016) [23]	Beth Israel Deaconess	1 Jan 2009–31 Dec 2013	319	147	172	MMG, US, MRI	Radiologist-defined MRI features	Analysis of variance, Regression	None			89%	83%
Tsukada (2022) [24]	Japan	Jan 2007– Jan 2012	51	19	32	MMG, US, DCE, PET/CT	BI-RADS, MRI, PET/CT SUV analysis	Logistic regression		92.30%		94.40%	73.00%
Others													
Kim (2020) [25]	Republic of Korea	Jul 2015– Jul 2018	105	31	74	ADC	ADC thresholding	Multivariate regression analysis	None	72%			
Thakur (2018) [26]	Memorial Sloan	Jan 2011–2013	31	10	21	ADC	ADC thresholding	Non-parametric Mann-Whitney’s test	None	78.5% (ADC600), 79.3% (ADC1000)			
Zhang (2021) [27]	America	2008–2010	80	34	46	Post-contrast BPE	K-means (Multiple post-contrast)	Mann-Whitney U test					
Ha (2019) [28]	Columbia U	Jan 2010– Jun 2016	134	17, 40	77	T1 post	Deep learning method	Convolutional Neural Network	5-fold	92% (3-class), 92% (2-class)	81% (3-class), 84% (2-class)	60% (3-class), 87% (2-class)	90% (3-class), 81% (2-class)

Footnote: All studies used ER+/HER2−, except Dialani (2016) [23] which had ER+ 100%, HER2− 95%, HER2+ 2%, HER2-equivocal 3%. Li (2016) [13] which had 19 HER2+. Note that performance metrics were those of external testing sets or internal testing sets, not training sets.

## Data Availability

Not applicable.

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
