# Peer review of "Radiomics Analysis of Breast MRI to Predict Oncotype Dx Recurrence Score: Systematic Review"

_diagnostics, 2025, doi:10.3390/diagnostics15091054_

Round 1

Reviewer 1 Report

Comments and Suggestions for Authors

This study is an interesting one, I agree with the authors message; Radiomics could serve as complementary tool, in particular ,for breast cancer patients with  intermediate or borderline ODXRS group.

I have only minor points to address:

1-      Line 60; Please expand the acronym (BPE) please once it is used in the manuscript for the first time.

2-      Line 63 : Please expand the acronym (ADC) please once it is used in the manuscript for the first time.

3-      Line 182: Association of other high markers as  Ki-67  with non-low ODXRS risk , suggest adding the following reference if possible :

  •           Saad Abdalla Al-Zawi A, Anichkina KA, Elamass M, Aladili Z. Correlation of Ki-67 proliferative index with oncotype DX recurrence score in hormone receptor-positive, human epidermal growth factor receptor 2-negative early breast cancer with low-burden axillary nodal disease - a review of 137 cases. Pol J Pathol. 2024;75(1):8-18. doi: 10.5114/pjp.2024.135859

4-      Line 351: Please expand the acronym (HIPAA) please once it is used in the manuscript for the first time.

Comments on the Quality of English Language

The quality of English is good, however it has minor language issues.

Author Response

This study is an interesting one, I agree with the authors message; Radiomics could serve as complementary tool, in particular ,for breast cancer patients with  intermediate or borderline ODXRS group.

I have only minor points to address:

1-      Line 60; Please expand the acronym (BPE) please once it is used in the manuscript for the first time.

2-      Line 63 : Please expand the acronym (ADC) please once it is used in the manuscript for the first time.

3-      Line 182: Association of other high markers as  Ki-67  with non-low ODXRS risk , suggest adding the following reference if possible :

  •           Saad Abdalla Al-Zawi A, Anichkina KA, Elamass M, Aladili Z. Correlation of Ki-67 proliferative index with oncotype DX recurrence score in hormone receptor-positive, human epidermal growth factor receptor 2-negative early breast cancer with low-burden axillary nodal disease - a review of 137 cases. Pol J Pathol. 2024;75(1):8-18. doi: 10.5114/pjp.2024.135859

4-      Line 351: Please expand the acronym (HIPAA) please once it is used in the manuscript for the first time.

Response: All of the above (1-4) have been addressed in the manuscript.

Reviewer 2 Report

Comments and Suggestions for Authors

The authors systematically reviewed 16 articles that included radiomix analyses of breast MRI in predicting the prognosis of breast cancer patients with Oncotype DX. It was a successful review.

The reasons and criteria for exclusion of articles identified in Pubmed should be added to the article.

Author Response

The authors systematically reviewed 16 articles that included radiomix analyses of breast MRI in predicting the prognosis of breast cancer patients with Oncotype DX. It was a successful review.

The reasons and criteria for exclusion of articles identified in Pubmed should be added to the article.

Response: The PRISMA Flowchart has been revised to show more detail regarding reasons for inclusion/exclusion of articles.

Reviewer 3 Report

Comments and Suggestions for Authors

Author Response

Thanks for giving the opportunity to read the paper entitled :”Radiomics analysis of breast MRI to predict Oncotype Dx recurrence score: Systematic review”. Generally, the radiomics study can be classified into two categories, one is handcrafted method, and the other is deep learning method. This review covers both topics and also includes survival analysis using Cox-regression method with radiomics.

It could be better if the authors organized the paper according to different machine learning methods or radiomic features extraction methods, pointing out the advantages and limitations of each study.

Response: We agree. Revised. We add the advantages and limitations of each group of methods and each study.

In the discussion section, including a paragraph to lay out the future research direction in this field.

Response: We agree. The following is added.

The number of references included in this review is relatively small.

Response: We agree. A few more are added.

There are other specific points in the following: Line 56: Radiomics analysis of MRI images have been used to predict the ODXRS ( add references here).

Response: Done

What is this “add references here”? Line 60: recurrence risk (REF). REF? I believe the authors would like to add a reference here, but forgot. Line 60: Quantitative assessment of BPE, what is BPE? Line 63: ADC: apparent diffusion coefficient? Find this abbreviation in line 156/202.

Response: Done

Table 1(page 4~6). It is not clear the AUC accuracy specificity and sensitivity were obtained from training, internal testing, or external testing data.

Response: Done

What were the methods for radiomic feature extraction? Did the authors adopt the IBSI features etc. (https://uk.mathworks.com/help/medical-imaging/ug/ibsi-standard-and-radiomics-functionfeature-correspondences.html )? How did the authors segment the tumour regions? Most of the study employed a few patients; only 3 studies have included more than 300 patients. Line 159: ADC600, ADC1000, did the authors used DWI with b value of 600 and 1000? Line 297: using ODXRS is enough here. Line 317/339: ROC has been used before, therefore, just use ROC. Same for AUC in line 319. Line 411: That is one of the reasons why researchers have developed IBSI standard, so that these radiomic features can be reproduced.

Response: Thank you for detailed reading of the manuscript. We agree your comments and provide the detailed if available. Note that these data are not available for all papers.

Reviewer comment:  It could be better if the authors organized the paper according to different machine learning methods or radiomic features extraction methods, pointing out the advantages and limitations of each study.

Response: Done.

Reviewer comments:  Radiomics analysis of MRI images have been used to predict the ODXRS (add references here).

AND

Line 60: recurrence risk (REF). REF? I believe the authors would like to add a reference here, but forgot.

Response: The 2 sentences are joined, and references re added.

Radiomics analysis of MRI images have been used to predict the ODXRS and these studies have shown that radiomics signatures derived from dynamic contrast-enhanced (DCE) MRI, T2-weighted imaging, and diffusion-weighted imaging (DWI) can be associated with ODXRS and recurrence risk [9-13].

Reviewer comment: Quantitative assessment of BPE, what is BPE? Line 63: ADC: apparent diffusion coefficient? Find this abbreviation in line 156/202. Table 1(page 4~6).

Response: These terms are now written out in their entirety when first mentioned, as per Reviewer 1’s recommendations.

Reviewer comment: Line 297: using ODXRS is enough here.

Response: Line number changed with revisions (Romeo et al.) but has been addressed and now states ODXRS.

Reviewer comment: Line 317/339: ROC has been used before, therefore, just use ROC.

Same for AUC in line 319.

Response: Line numbers changed with revisions, all under Chiacciaretta et al. All have been addressed.

Reviewer comment: Line 411: That is one of the reasons why researchers have developed IBSI standard, so that these radiomic features can be reproduced.

Response: 1st sentence of second paragraph under section 4.3.

Reviewer comment:

It is not clear the AUC accuracy specificity and sensitivity were obtained from training, internal testing, or external testing data. What were the methods for radiomic feature extraction? Did the authors adopt the IBSI features etc. (https://uk.mathworks.com/help/medical-imaging/ug/ibsi-standard-and-radiomics-functionfeature-correspondences.html )?

Response: They varied. We tabulated performance metrics for external testing data (if available) or internal testing. We did not report training performance metrics. Methods used for radiomic feature extraction are many and heterogeneous, not all adopt IBSI features. We could not tabulate in the table but describe them in text.

Note that performance metrics were those of external testing sets or internal testing sets, not training sets.

How did the authors segment the tumour regions? Most of the study employed a few patients; only 3 studies have included more than 300 patients.

Response: Different automated and manual segmentation methods of tumor were used. We could not tabulate in the table but describe them in text if available.

Line 159: ADC600, ADC1000, did the authors used DWI with b value of 600 and 1000?

Response: That is correct.

Reviewer 4 Report

Comments and Suggestions for Authors

This systematic review has made a valuable contribution to the field of predicting Oncotype DX recurrence scores using radiomics analysis of breast MRI, providing important references and directions for future research. However, to further enhance the quality and credibility of the review, the authors are advised to consider the following points in their revision:The authors should include an assessment of the quality of the included studies, clearly define the criteria and methods for quality evaluation, and discuss the impact of study quality on the conclusions in the results section.The authors should delve deeper into the sources of heterogeneity among the studies and employ appropriate methods to address or reduce this heterogeneity in order to enhance the reliability and comparability of the results.The authors are encouraged to conduct a more in-depth analysis of the feasibility and challenges of applying radiomics models in clinical settings, including aspects such as technology, resources, ethics, and patient acceptance.The authors should supplement the discussion section with ethical considerations regarding patient privacy and data security to ensure that the research complies with ethical standards and provides guidance for clinical application. If the authors can address these issues in their revision, I believe that this paper will become more persuasive and practically valuable, offering new insights and methods for the assessment of breast cancer recurrence risk.

Comments on the Quality of English Language

The English could be improved to more clearly express the research.

Author Response

This systematic review has made a valuable contribution to the field of predicting Oncotype DX recurrence scores using radiomics analysis of breast MRI, providing important references and directions for future research. However, to further enhance the quality and credibility of the review, the authors are advised to consider the following points in their revision:

The authors are encouraged to conduct a more in-depth analysis of the feasibility and challenges of applying radiomics models in clinical settings, including aspects such as technology, resources, ethics, and patient acceptance. The authors should supplement the discussion section with ethical considerations regarding patient privacy and data security to ensure that the research complies with ethical standards and provides guidance for clinical application.

Response: The following is added.

Applying radiomics models in clinical settings affords several advantages, but also requires overcoming challenges across multiple domains, including technology, resources, ethics, and patient acceptance. One of the major challenges in applying radiomics models in clinical settings is the lack of standardization in imaging protocols. Scanners by different manufacturers and variations in scanning techniques leads to variability in data, which complicates the application of a uniform radiomics model. There are therefore inherent challenges to model generalizability. In addition, radiomics analysis is computationally intense and requires sufficient resources to accomplish its tasks. This requires a highly sophisticated infrastructure, including servers and AI with sufficient capacity to achieve real-time radiomics analysis, not to mention the increased costs involved for hardware and software acquisitions, training of professionals, and computational power. These challenges are a significant barrier in resource-limited settings.  Integration into clinical workflow would also require that ethical concerns be addressed. While radiomics may improve diagnostic accuracy and advance personalized treatment, data sharing and privacy concerns, compliance with ethical standards, risk of algorithm bias, patient consent and informed decision-making are issues that would need to be addressed. The “black box” nature of machine learning algorithms makes it difficult to apply in the real world, as clinicians can be given no explanation as to how the algorithm arrives at a decision, making it difficult to establish trust. Strategies to overcome some of these issues include strict adherence to data protection laws, testing and validation of radiomics models across diverse datasets and clinical environments to improve their generalizability, and standardization of imaging protocols, model design, and feature extraction techniques. Interdisciplinary collaboration between radiologists, oncologists, data scientists, and ethicists can help develop and implement radiomics solutions that are both clinically effective and ethically sound. Transparency in communicating to patients the use of their data, benefits of radiomics models, and their rights to privacy and data security can help to establish trust and gain patient acceptance. Addressing these challenges would make the clinical application of radiomics a feasible and desirable resource to add to the clinician’s toolbox to improve diagnostic accuracy, promote personalized treatment plans, and ultimately improve patient outcomes.

The authors should include an assessment of the quality of the included studies, clearly define the criteria and methods for quality evaluation, and discuss the impact of study quality on the conclusions in the results section. The authors should delve deeper into the sources of heterogeneity among the studies and employ appropriate methods to address or reduce this heterogeneity in order to enhance the reliability and comparability of the results. If the authors can address these issues in their revision, I believe that this paper will become more persuasive and practically valuable, offering new insights and methods for the assessment of breast cancer recurrence risk.

Response: The following is added.

Some sources of heterogeneity among studies includes use of different machines, protocols, and technical parameters, making it difficult to achieve uniform results. This can be challenging with multi-institutional studies involve varied protocols across multiple institutions which affects uniformity of study design and may affect results. Li et al. attempted to mitigate this problem by limiting their inclusion to patients who were scanned on 1.5 T GE Medical systems imaging units (Milwaukee, Wis) from 4 different institutions. The process and methods of segmentation also introduces variability. While most studies utilized some method of automated segmentation for radiomics analysis, one study used purely radiologist drawn ROI, which is work-intensive [15]. Sample size is also important to achieve prognostic accuracy in radiomics. Only 3 of the studies included more than 300 patients [21,21, 23]. This can be problematic when many features are extracted from a small sample size, which may lead to overfitting and limit generalizability. Most of the studies were from a single institution, limiting generalizability. While texture features can be useful, radiomics extracts a broader range of data, including tumor shape, size, and post-contrast enhancement kinetics, which may improve prediction analyses. Multiparametric radiomics exploits the advantages of radiomics and multiple types of image acquisitions (DCE, T2, DWI), which yielded the highest AUC (92%) among the studies [16]. The use of a deep learning CNN algorithm was able to predict ODXR with a similarly high AUC of 92% [28].

Round 2

Reviewer 4 Report

Comments and Suggestions for Authors

I have no other problems